# Developed Design of Battle Royale Optimizer for the Optimum Identification of Solid Oxide Fuel Cell

Keyvan Karamnejadi Azar [1], Armin Kakouee [2], Morteza Mollajafari [3], Ali Majdi [4], Noradin Ghadimi [5,6,*] and Mojtaba Ghadamyari [7,8]

1  Department of Electrical Engineering, Urmia Branch, Islamic Azad University, Urmia 57169-63896, Iran
2  Department of Mechanical Engineering, Amoli Branch, Islamic Azad University Ayatollah, Amol 46351-43358, Iran
3  Automotive Electrical and Electronics Laboratory, School of Automotive Engineering, Iran University of Science and Technology, Tehran 16846-13114, Iran
4  Department of Building and Construction Techniques, Al-Mustaqbal University College, Hillah 51001, Iraq
5  Young Researchers and Elite Club, Ardabil Branch, Islamic Azad University, Ardabil 56157-31567, Iran
6  Department of Industrial Engineering, Ankara Yıldırım Beyazıt University, Ankara 06760, Turkey
7  Department of Computer Engineering, Lebanese French University, Erbil 44001, Iraq
8  Department of Electrical Engineering, Shahid Beheshti University, Tehran 19839-69411, Iran
*  Correspondence: nghadimi@ybu.edu.tr

**Abstract:** One of the most appropriate electricity production systems is solid oxide fuel cells (SOFCs), which are important because they are highly efficient, flexible to fuel, and have fewer environmental degradation effects. A new optimum technique has been provided for providing well-organized unknown parameters identification of the solid oxide fuel cell system. The main idea is to achieve the lowest amount of the sum of square error between the model's output voltage and the empirical voltage datapoints. To get efficient results, the minimum error value has been achieved by designing a new metaheuristic algorithm, called the Developed version of Battle Royale algorithm. The reason for using this version of Battle Royale algorithm is to achieve results with higher accuracy and better convergence. The proposed technique was then applied to a 96-cell SOFC stack under different temperature and pressure values and its achievements were compared with several different latest methods to show the proposed method's efficiency.

**Keywords:** solid oxide fuel cell; parameter estimation; Developed Battle Royale algorithm; sensitivity analysis

## 1. Introduction

Humanity is significantly dependent on energy. There is a deep gap between consumers and fossil fuel accessibility due to the increase in economic growth and social development [1]. Today, the energy crisis and environmental pollution caused by fossil fuels is a major problem [2,3]. Increasing human activities and the use of natural energy sources has caused a reduction in clean energy, and the methods currently in place to produce energy are not environmentally friendly [4]. Worries about global climate warming require the development of new techniques of energy generation by natural sources of carbon and sustainable energy [5,6]. Different clean energy sources have been designed and utilized to solve this issue [5]. For example, photovoltaic systems, wind turbines, and fuel cells (FCs) are some renowned sources in this area [7]. Among them, fuel cells, as a new and environmentally friendly technology, have become more studied and utilized [8]. Fuel cells are a technology with high efficiency for converting chemical energy into electrical energy [9,10]. Hydrogen is used as the cleanest energy as fuel in fuel cells [11]. Hydrogen is one of the most abundant elements on Earth [12]. This element does not exist in nature in its pure form, but it can be obtained in several different ways from other elements [13]. Due to the depletion of oil reserves and pollution of fossil fuels, the use of fuel cells is necessary

in the future which does not pollute the environment in cars and power plants. Fuel cells have several advantages, for example,

- They do not produce $H_2$ fuel, greenhouse gas, or pollution of air.
- They significantly cause environment enhancement [14].
- They are more efficient than combustion engines.
- Unlike co-generation uses, these cells generate heat and electrical power with efficiency of about 80%.
- FCs generate water and heat without particles, GHGs, or toxins, i.e., these cells generate unpolluted air.
- They can be used in various sizes from mWs to MWs, such as in buildings, mobile phones, cars, etc.
- FC supplements are applicable in various energy techniques, such as wind turbines, batteries, solar panels, and super capacitors [8].

There are several models of fuel cells that are divided based on their configurations. Each of these types also have their advantages and disadvantages [15]. Recently, the utilization of solid oxide fuel cells (SOFCs) in high-temperature systems, such as power plants, has made them devices with high tendency from the industry society [16]. Because of high temperature working points, designing and testing these models needs a lot of time and comes with a high cost. A proper method to prevent these problems is to first model and simulate these devices in the computer and after reaching to the best efficiency, they can be designed and constructed [17]. Another problem is that the manufactured SOFCs after sending to the costumers, have some unknown parameters in their model which should be then selected optimally for the considered work. Due to the high cost of polymer electrolytes, it is better to predesign and simulate it. Some papers are published in this regard [18]. However, literature showed that using metaheuristic-based techniques can be so useful for solving such complicated problems. Different metaheuristics are proposed in this direction. For example, Improved Red Fox Optimizer (IRFO) [19], Marine Predator Optimizer (MPO) [20], chaotic grey wolf metaheuristic algorithm [21], Levenberg–Marquardt republishing optimization algorithm [22], genetic algorithm, and radial movement optimization [23] are some of these techniques. In the following, more details of the algorithms are explained.

Luo et al. [19] used metaheuristic methods to detect solid oxide fuel cell (SOFC) parameters. In this research, an optimized metaheuristic method is used to detect the parameters of SOFC. The optimized metaheuristic method is called Improved Red Fox Optimizer (IRFO). For the evaluation of the effectiveness, an optimized metaheuristic model used the Sum of Squared Error (SSE). The finding displayed that the IRFO method had minimum error to generate power in various temperature conditions. It was obtained by about 0.0073 kW. Moreover, the optimized metaheuristic method had a minimum error value for output voltage by about 0.16 V. This finding was acceptable to detect SOFC parameters.

Yousri et al. [20] detected unknown parameters of SOFCs dynamically simulating by comprehensive learning dynamic multiple-swarm MPO method. The most important step in presenting an energy storage system is to accurately identify the unknown parameters. In this research, with the help of the Marine Predator Optimizer (MPO), the proposed strategies detected the variables. The efficiency of the proposed optimization method was evaluated in different conditions of sudden load changes and dynamic voltage responses. Comparison of the results showed that the Marine Predator Optimizer (MPO), by the suggested strategies had the highest accuracy in identifying unknown parameters and presented the lowest variance between current and voltage.

Hao et al. [21] utilized the improved chaotic grey wolf metaheuristic algorithm to detect solid oxide fuel cells variables. The purpose of their use of optimization methods is to increase the speed and accuracy of identifying the parameters of solid oxide fuel cells. The results of using the chaotic grey wolf optimization method showed that this optimization method had the least mean square error, the highest accuracy, and stability in the optimal identification of unknown parameters. It also had the highest convergence

rate for resolving optimization problems. Moreover, it was able to provide the minimum variance between the current and voltage curves.

Yang et al. [22] identified the unknown parameters of the SOFC using the metaheuristic method to identify SOFC parameters for modeling. For this objective in this study, the Levenberg–Marquardt republishing optimization algorithm was used. This algorithm was used to increase the efficiency of the artificial neural network in the training process to identify unknown parameters. The results suggested by the technique were compared with the electrochemical model (ECM) and steady-state model (SSM). The optimized artificial neural network had good simulation results in identifying the parameters so that this improved model was able to identify unknown parameters with high speed and stability and provide more accurate results.

Nassef et al. [23] applied an optimal ANN method to identify the accuracy of solid oxide fuel cell parameters. The optimal ANN model was based on radial movement optimization. To confirm the optimum artificial neural network, it was compared to a genetic algorithm. For this objective, the genetic algorithm and radial movement optimization were compared in two different conditions. The investigation methods showed that the radial movement optimization method increased the power by about 17.28% compared to the genetic optimization algorithm. Furthermore, in the second condition, the radial movement metaheuristic method increased the power compared to the genetic method about 28.85%.

Based on the literature, efficient results can be provided by optimization algorithms for system identification of the SOFCs. Likewise, the application of metaheuristics as a famous part of optimizers is continuously being increased. It is worth noting that by the "No free lunch" theorems [24], there is no metaheuristic algorithm with the capability of solving any type of problem. The major target here is to propose a developed design of a metaheuristic optimizer to deliver a SOFC model with higher efficiency. Therefore, the main contributions of the present study can be briefly highlighted as follows:

- New optimal parameters estimation of the solid oxide fuel cell system based on metaheuristics.
- The idea is to minimize the error between the model output and the empirical datapoints.
- A developed version of Battle Royale algorithm is utilized to minimize the error value.
- The method is performed on a 96-cell SOFC stack under different temperature and pressure values.

## 2. Modeling of a SOFC

The operational temperature of SOFC ranges between 600 °C and 1000 °C and therefore various kinds of fuels can be applied in it. This fuel cell has two plate and tube structures and a thin ceramic solid electrolyte is used instead of a liquid electrolyte. At high operating temperatures in a solid oxide fuel cell, oxygen ions (negatively charged) move over the crystal lattice [25]. At the negative electrode, four electrons have been combined with $O_2$. While a hydrogen-containing gas fuel moves over the positive electrode, $O_2$ ions with negative load moves through the electrolyte for fuel oxidization [26]. The generated electrons at the positive electrode create an exterior circuit to produce electricity. SOFCs do not require an additional converter. Solid oxide combustion reactions are given below [27]:

For cathode side:

$$2H_2 + 2O^{2-} \rightarrow 2H_2O + 4e^- \tag{1}$$

For anode size:

$$O_2 + 4e^+ \rightarrow 2O^{2-} \tag{2}$$

And the total reaction of the system is:

$$2H_2 + O_2 \rightarrow 2H_2O \tag{3}$$

The efficiency of SOFC is almost 45–60% and the density of power generation in this type of fuel cell is 240–300. The lifespan of it is more than 40,000 h. The total efficiency is

increased to 70% by combination of this cell with a turbine. Figure 1 shows a general form of a SOFC.

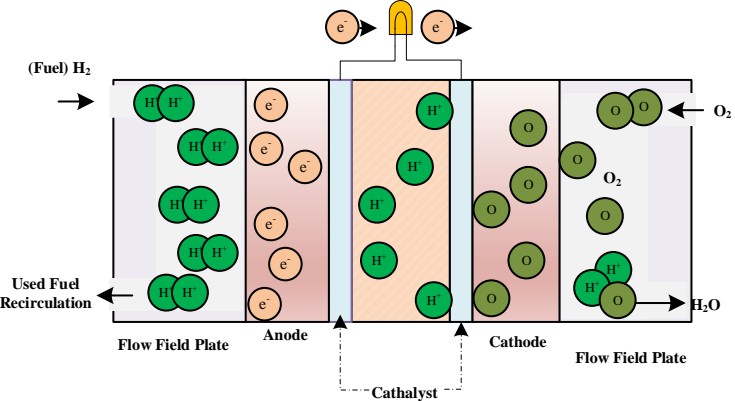

**Figure 1.** General form of a SOFC.

As can be observed from Figure 1, by considering the high operating temperature operation condition, the oxygen ions, which are charged negatively, move through membrane which is typically a blend of calcium oxide and zirconium oxide. The $O_2$ molecules are then combined with 4 electrons at the cathode side. During the process, if a gassy fuel containing hydrogen molecules moves through the anode, there will be a movement by the negatively charged current including oxygen ions moving over the electrolyte for fuel oxidization. The electrons that are generated at the positive electrode side move over an exterior circuit to reach to the negative electrode side and produces electricity.

By considering the configuration explained before, the SOFC generate electricity with $V_T$ voltage. This voltage is achieved by considering different voltage drops that have happened on the fuel cell. The main voltage losses in the SOFC include activation voltage drop, concentration voltage drops, and ohmic voltage drop. By considering these losses, the outputted voltage of the SOFC is achieved as follows [28]:

$$V_T = (E_N - V_\Omega - V_{Cons} - V_{Act}) \times N \tag{4}$$

where $V_\Omega$ describes the ohmic voltage drop, $V_{act}$ defines the activation voltage, and $V_C$ specifies the concentration voltage drop, $N$ determines the quantity of the cells, and $E_N$ is the open circuit voltage which is achieved by the following formula:

$$E_N = 2 \times F \times E_0 + R \times ln\left(P_{H_2} \times \sqrt{P_{O_2}} \times 1/P_{H_2O}\right) \times T \times \left(\frac{0.5 \times N_0}{F}\right) \tag{5}$$

where $E_0$ describes the reversible potentiality, and $T$ specifies the operational temperature. Table 1 reports the parameter value of the parameters.

**Table 1.** The parameter amounts of the determined SOFC parameters.

| Parameter | Value |
|:---:|:---:|
| $T$ | $1.253\text{–}2.4516 \times 10^{-4}$ |
| $F$ | $96,486 \text{ C mol}^{-1}$ |
| $R$ | $8.314 \text{ kJ (kmol K)}^{-1}$ |

$P_{O_2}$, $P_{H_2}$, and $P_{H_2O}$ describe the partial pressure for $O_2$, $H_2$, and the $H_2O$, respectively. The partial pressure for Oxygen and Hydrogen are achieved by the following [29]:

$$P_{O_2} = R_{hc} \times P_{H_2O} \times \left[\frac{P_c}{R_{hc} \times P_{H_2O} \times e^{\frac{1.635I/A}{T^{1.334}I}}} - 1\right] \tag{6}$$

$$P_{H_2} = 0.5 \times R_{ha} \times P_{H_2O} \times \left[ \frac{P_a}{R_{ha} \times P_{H_2O} \times e^{\frac{1.635 I/A}{T^{1.334} I}}} - 1 \right] \qquad (7)$$

where $P_a$ and $P_c$ represent the inlet pressure in anode and cathode, and $R_{ha}$ and $R_{hc}$ represent the relative humidity of vapor at the positive and negative electrodes sides.

The ohmic voltage loss of the SOFC can be calculated as follows:

$$V_\Omega = I \times r \qquad (8)$$

And the concentration voltage loss for the cell is mathematically achieved as follows:

$$V_C = \frac{R \times T \times \left[ \ln\left( P_{H_2}^2 \times P_{O_2} \times P_{H_2O}^{-2} \right) - \ln\left( P_{H_2}^{*2} \times P_{O_2}^{*} \times P_{H_2O}^{*2} \right) \right]}{4 \times F} \qquad (9)$$

where $P_{H_2}^{*}$ defines the partial pressure of the hydrogen, $P_{O_2}^{*}$ partial pressure of oxygen, and $P_{H_2O}^{*}$ partial pressure of water.

The activation voltage drop is defined as given below:

$$V_A = C_T + S \times \ln I \qquad (10)$$

where $C_T$ represents the Tafel coefficient and $S$ is the slope that can be formulated as follows:

$$C_T = \left( -\frac{RT}{\alpha \times n_e \times F} \right) \qquad (11)$$

$$S = \left( \frac{RT}{\alpha \times n_e \times F} \right) \qquad (12)$$

Finally, the terminal voltage of a SOFC can be obtained as below:

$$V_T = E_0 - I \times R_\Omega - A \times \sinh^{-1}\left( \frac{I}{2 \times I_0^a} \right) - A \sinh^{-1}\left( \frac{I}{2 \times I_0^c} \right) + B \times \ln\left( \frac{I_L - L}{I_L} \right) \qquad (13)$$

where $R_\Omega$ signifies the resistance of the device area (k$\Omega$ cm$^2$), $I_L$ specifies the constraint of current density (mA cm$^2$), and $I_0^c$ and $I_0^a$ refer to the exchanging flow's current density of the cathode and anode, respectively. By considering the clarified equations, seven unknown parameters are defined for optimization. The parameters include $R_\Omega$, $A$, $B$, $E_0$, $I_L$, $I_0^a$, and $I_0^c$.

## 3. Objective Function

As aforementioned, the major target here is to provide an optimum parameter estimation methodology for the undetermined parameters of the SOFC models. The concept is to lessen the means square error (MSE) between the experimental data and the model output data. The data in this study are the voltage profile. Thus, the best results will be achieved when there is a high confirmation between the empirical data and the model output voltage. This conception can be considered as an objective function as follows [30]:

$$OF = \frac{1}{N} \sum_{i=1}^{N} \left( V_T(i) - V_{exp}(i) \right)^2 \qquad (14)$$

where $N$ specifies the sample number of the voltage data, $V_T$ and $V_{exp}$ describe the terminal and the experimental output voltage data, respectively. So, the main idea is:

$$CF = minOF \qquad (15)$$

Considering the undetermined variables, $R_\Omega$, $A$, $B$, $E_0$, $I_L$, $I_0^a$, and $I_0^c$, as decision variables subject to the following constraints [31]:

$$I \leq I_L \qquad (16)$$

$$I_0^c < I_0^a \tag{17}$$

$$\underline{x}_i \leq x_i \leq \overline{x}_i \tag{18}$$

Since the defined function in Equation (15) is a nonlinear formulation, solving it with classic methods can sometimes provide improper results. In this study, we propose an improved metaheuristic-based methodology for minimizing Equation (15).

## 4. Battle Royale Optimization Algorithm (DBRA)

### 4.1. Strategy of Battle Royale Game

Battle Royale is an enduring and competitive game designed using a Japanese movie. In this game, players must explore the game area to avoid being eliminated. Battle Royale can be played both in pairs and in teams of up to five people [32]. The players of this game have the same power and resources because beating is one of the challenges of the game. In addition, at the beginning of the game, all the players are randomly distributed in the game space. The game space gradually decreases and if a player leaves the game range (safe zone), he will either be expelled from the game or will be injured. Players must explore tools to stay in the game in a smaller playing area [33]. In some kinds of Battle Royale games, such as Counter-strike: Global Offensive, Ring of Elysium, Apex Legends, Player Unknown's Battlegrounds (PUBG), and Call of Duty: Warzone, players are given the opportunity to rejuvenate; this feature is used in the proposed algorithm, and sometimes players are rewarded for staying in the game [34]. In the end, only one team or one player wins. In the mentioned games, the map of the game area is selected by the players, for example, one of the famous maps in the PlayerUnknown's Battlegrounds is Sanhok. Players outside the safe zone of the game, which is also called "the circle", are injured and damage ticks are sent to them, so players try to be near to each other during the game, the map of this area is marked in blue. Over time, the safe area becomes smaller, and the next limited space is marked with a small white circle. Players try to kill their rivals to continue the game themselves. One of the types of the game of the PlayerUnknown's Battlegrounds is that players kill or slice a certain number of rivals at a set time. This is called a death match mode, and the player who kills more rivals will be winner. Furthermore, during the game, the player who is killed can appear randomly in the game area.

### 4.2. Battle Royale Optimization Algorithm

In some cases, players jump from a parachute or a plane down onto the map and the game starts. Battle Royale Optimization Algorithm is population-based and the initial population is evenly dispersed all over the research space. Each soldier according to the location he is in shoots the soldiers around him and injures them, and the level of damage to the injured soldier increases to one and is expressed by damage = $z_j$.damage + 1. The injured soldier wants to alter his location immediately and shoot at the rivals from the other flank. The soldier is placed between the preceding location and the best location found so that he can focus on exploitation. The mathematical expression of these behaviors is presented below [33]:

$$z_{dam.d} = z_{dam.d} + r(z_{best.d} - z_{dam.d}) \tag{19}$$

The location of the injured soldier is shown by $z_{dam.d}$ which is in dimension $d$, $r$ is a random amount in the range [0, 1]. In the following iteration, if the injured soldier can shoot the rival, the value of $z_j$ is reduced to zero. In the exploration phase, if the amount of damage of $j$th soldier ($z_j$) exceeds the pre-determined threshold amount, it dies but can appear randomly in the game space, and $z_j$.damage can reduce to 0. After trial and error, the amount of threshold is set on three. This prevents premature convergence and creates good exploration. A soldier who appears in the search space after being killed is expressed by the following formula [33]:

$$z_{dam.d} = r \times (ub_d - lb_d) + lb_d \tag{20}$$

The upper and lower bounds in problem space are indicated by $ub_d$ and $lb_d$. The initial amount of problem space is $\omega = \log_{10}(MaxCicle)$ with each epoch, the search space of the problem becomes smaller according to the best solution and its value is $\omega = \omega + round\left(\frac{\omega}{2}\right)$. Maximum amount of generation is expressed by *MaxCicle*.

The updated upper and lower bound values are given below [33]:

$$lb_d = z_{best.d} - SD(\overline{z_d})$$
$$ub_d = z_{best.d} + SD(\overline{z_d}) \quad (21)$$

where the best solution ever found in dimension *d* is shown by $z_{best.d}$, $SD(\overline{z_d})$ is the standard deviation of entire population. If the lower/upper bound exceeds the original $lb_d/ub_d$, it sets to the original lower/upper bound. The finest soldier in each epoch is saved and considered as an elite.

The computational difficulty of the suggested algorithm depends on the dimensions of the problem, the highest number of epochs, and the population number. According to the number of populations $(n)$ and the number of epochs $(m)$, the computational complexity for all solutions is $O(n^2)$ and $O(m^2)$, respectively.

### 4.3. Developed Battle Royal Optimization Algorithm

However, the Battle Royal Optimization Algorithm as a new metaheuristic approach provides good results based on its defaults analysis [35], it may have weak results in some problems in different terms such as improper convergence and local optimum results [36]. Therefore, here, we used two kinds of modifications to improve the algorithm efficiency for designing a more efficient model estimator for the SOFC mathematical model [37]. The first improvement is based on the opposition-based learning (OBL) mechanism. Based on the OBL mechanism [38], each generated solution candidate is considered as a pair of candidate, where the place of the candidates pair is complement of the main candidate, i.e., [38],

$$\overrightarrow{z}_j^{\text{new}} = \overrightarrow{z}_j^{max} + \overrightarrow{z}_j^{min} - \overrightarrow{z}_j \quad (22)$$

where $\overrightarrow{z}_j^{\text{new}}$ describes the opposite position of $\overrightarrow{z}_j$, and $\overrightarrow{z}_j^{min}$ define the minimum and $\overrightarrow{z}_j^{max}$ is the higher limitations of the solution.

The better solution of each pair will be considered as the new candidate and the other one will be removed.

Here, 60% of the initial population is achieved randomly and 40% is achieved based on the OBL mechanism. The other improvement that is used herein is "chaos theory". This system generates pseudo-random variables instead of completely random variables. This can be used in the metaheuristics to enhance the algorithm's convergence speed. Different chaos introduced in the literature. This study uses a sinusoidal map for this purpose. By updating the *r* parameter in Equation (20) as a pseudo-random variable, its update formula will be achieved as follows [39]:

$$r_1(j+1) = P \times r_1^2(j) \times \sin(\pi \times r_1(j)) \quad (23)$$

where $r_1(j+1)$ specifies the chaotic random value generated during the current iteration, and $r_1(j)$ defines the chaotic random value generated in the preceding iteration. Here, $P = 2.2$ is the control parameter and $r_1(0)$ has been set 0.6.

The workflow of the Developed Battle Royal Optimizer is given in Figure 2.

To provide a proper validation to evaluate the suggested Developed Battle Royal Algorithm efficiency, the algorithm was performed to several standard test functions and its achievements were compared with the Particle Swarm Optimizer (PSO) [40] and two of the newest presented optimizers: Whale Optimizer (WO) [41] and Archimedes Optimizer (AO) [42]. Table 2 states the controlling parameters of the investigated algorithms.

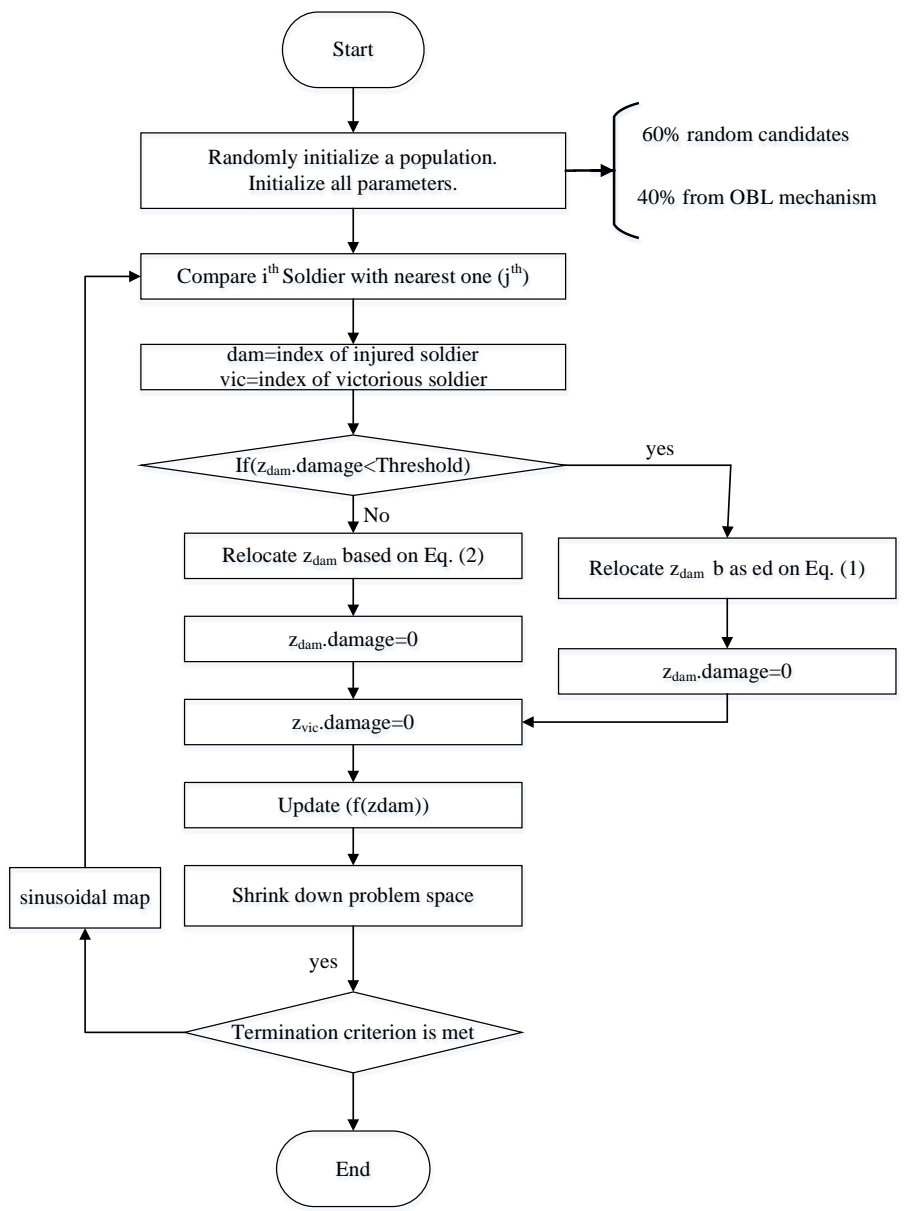

**Figure 2.** Developed Battle Royal Algorithm.

**Table 2.** Controlling parameters of the investigated algorithms.

| Algorithm | Parameter | Value |
|---|---|---|
| Particle Swarm Optimization (PSO) [40] | $c_1$ and $c_2$ | 1 |
| | $w$ | 0.7 |
| Whale Optimization Algorithm (WO) [41] | $\vec{a}$ | 2 |
| | $\vec{r}$ | 1 |
| Archimedes Optimization Algorithm (AO) [42] | Protection probability | 10% |
| | Elimination probability | 25% |
| | $c_1$ | 1.5 |
| | $c_2$ | 1.5 |

The simulation was performed in MATLAB R2018b software and programmed on a Core i7-CPU 2.00 GHz processor with 16 GB of RAM. Four mathematical test functions are used for comparison [43–46]. The information about these functions is listed in Table 3.

**Table 3.** Information about the utilized benchmark functions.

| Formulation | Range | $F^*$ |
|---|---|---|
| $F1 = x \times \sin(4x) + 1.1y \times \sin(2y)$ | $0 < x,\ y < 0$ | $-18.55$ |
| $F2 = 0.5 + \frac{\sin^2\left(\sqrt{x^2+y^2}-0.5\right)}{1+0.1(x^2+y^2)}$ | $0 < x,$ $y < 2$ | $0.5$ |
| $F3 = \|x\| + \|y\| + \left(x^2 + y^2\right)^{0.25} \times \sin\left(30((x + 0.5)^2 + y^2)^{0.1}\right)$ | $[-\infty, \infty]$ | $-0.25$ |
| $F4 = 10n + \sum\limits_{i=1}^{n}\left(x_i^2 - 10\cos(2\pi x_i)\right),\quad n = 9$ | $[-5.12, 5.12]$ | $0$ |

The size of the population was 60 and highest iterations number during the optimization was 100. Because of the stochastic nature of the metaheuristic algorithms, all of the empirical achievements were achieved by considering the mean value of applying 35 runs independently for all of the benchmark functions. Moreover, the average value, min value, max value, and standard deviation (Std) of the test functions after 35 runs independently were used as the measurement indicators. Table 4 reports the numerical achievements of the optimizers employed to the investigated test functions.

**Table 4.** Numerical achievements of the optimizers employed to the investigated test functions.

| Function | Indicator | Algorithm | | | |
|---|---|---|---|---|---|
| | | PSO [40] | WO [41] | AO [42] | DBRA |
| $F_1$ | Max | $-11.232$ | $-14.012$ | $-16.263$ | $-19.64$ |
| | Min | $-15.287$ | $-15.646$ | $-14.41$ | $-14.33$ |
| | Median | $-13.253$ | $-14.558$ | $-15.16$ | $-16.54$ |
| | Std | $5.355$ | $4.839$ | $3.64$ | $2.34$ |
| $F_2$ | Max | $0.647$ | $0.637$ | $0.325$ | $0.315$ |
| | Min | $0.453$ | $0.427$ | $0.413$ | $0.4$ |
| | Median | $0.55$ | $0.537$ | $0.48$ | $0.476$ |
| | Std | $0.038$ | $0.012$ | $0.003$ | $0.001$ |
| $F_3$ | Max | $-0.089$ | $-0.187$ | $-0.213$ | $-0.221$ |
| | Min | $-0.212$ | $-0.234$ | $-0.246$ | $-0.297$ |
| | Median | $-0.150$ | $-0.164$ | $-0.210$ | $-0.263$ |
| | Std | $0.134$ | $0.108$ | $0.018$ | $0.013$ |
| $F_4$ | Max | $15.363$ | $12.437$ | $9.254$ | $2.374$ |
| | Min | $1.816$ | $1.009$ | $0.008$ | $0.002$ |
| | Median | $8.589$ | $6.721$ | $4.634$ | $1.189$ |
| | Std | $5.234$ | $5.054$ | $2.372$ | $1.062$ |

Based on Table 4, the suggested Developed Battle Royal Optimizer in all four analyzed benchmark functions provided the best results with minimum value that indicates its higher effectiveness toward the comparative optimizers. Furthermore, observed from the achievements, the suggested technique gives the minimum standard deviation results that show its higher reliability in solving the methods during different independent runs.

## 5. Simulation Results

As can be observed from the previous sections, the major target of this paper is to provide an efficient method for system identification of the SOFCs. The major concept is to provide an optimum method by a developed design of Battle Royal Optimizer to minimize the sum of square error between the experimental data and the estimated outputted voltage from the designed system. The model is based on working on the mathematical conception of the SOFC with considering some undetermined variables on the model as decision values which should be optimally selected for minimizing the error value. The decision variables are $R_\Omega$, $A$, $B$, $E_0$, $I_L$, $I_0^a$, and $I_0^c$. As mentioned before, due to using the stochastic algorithm here, we ran the algorithm 25 times independently and its mean value was considered as the final solution.

For ability validation of the presented Developed Battle Royal Optimizer in optimal parameter selection of the SOFC system, it was implemented to a studied case and its achievements were compared with some other related latest techniques by Chaotic Gray Wolf Optimization algorithm (CGWO) [47], Satin Bowerbird Optimizer (SBO) [48], Simplified Competitive Swarm Optimizer (SCSO) [49], and Teaching-Learning Based Algorithm (TLBO) [50]. To provide a fair comparison, like the Developed Battle Royal algorithm, all of the compared methods were run 25 times and the maximum iteration and the population size for all of them were similarly set to 200 and 50, respectively.

The case study in this research is a 96-cell SOFC stack where its information was collected from [51]. The minimum and higher constraints of the major parameters for the studied case is given in Table 5.

**Table 5.** Lower and upper limitations of the main parameters for the investigated case.

| Parameter | Lower Bound | Higher Bound | Unit |
|:---:|:---:|:---:|:---:|
| $E_{OC}$ | 0 | 1.2 | V |
| $A$ | 0 | 1 | V |
| $B$ | 0 | 1 | V |
| $I_L$ | 0 | 10,000 | mA·cm$^{-2}$ |
| $I_{0,a}$ | 0 | 100 | mA·cm$^{-2}$ |
| $I_{0,c}$ | 0 | 1 | mA·cm$^{-2}$ |

In the first step, the system efficiency was analyzed based on temperature variations under constant pressure. Therefore, this assessment shows how temperature changing can affect the identification system. Here, we used 160 datapoints for the analysis. In this study, five different temperatures, namely 550 °C, 600 °C, 650 °C, 700 °C, and 750 °C, under 3 atm constant pressure value were analyzed. Table 6 reports the simulation achievements of the suggested technique under different temperatures in comparison with some of the latest algorithms.

**Table 6.** Simulation results of the suggested technique under different temperatures in comparison with some of the latest algorithms.

| Algorithms | 550 °C | 600 °C | 650 °C | 700 °C | 750 °C |
|:---:|:---:|:---:|:---:|:---:|:---:|
| CGWO [47] | $6.28 \times 10^{-2}$ | $7.23 \times 10^{-2}$ | $9.92 \times 10^{-2}$ | $2.19 \times 10^{-1}$ | $6.63 \times 10^{-1}$ |
| SBO [48] | $4.15 \times 10^{-2}$ | $5.09 \times 10^{-2}$ | $7.16 \times 10^{-2}$ | $8.80 \times 10^{-2}$ | $2.98 \times 10^{-1}$ |
| SCSO [49] | $8.16 \times 10^{-3}$ | $9.11 \times 10^{-3}$ | $2.09 \times 10^{-2}$ | $5.25 \times 10^{-2}$ | $7.46 \times 10^{-2}$ |
| TLBO [50] | $5.50 \times 10^{-3}$ | $7.09 \times 10^{-3}$ | $9.05 \times 10^{-3}$ | $1.39 \times 10^{-2}$ | $4.28 \times 10^{-2}$ |
| DBRA | $9.41 \times 10^{-5}$ | $6.39 \times 10^{-4}$ | $8.60 \times 10^{-4}$ | $1.63 \times 10^{-3}$ | $3.97 \times 10^{-3}$ |

As can be observed from Table 6, the proposed DBRA with $9.41 \times 10^{-5}$ in lower temperature, i.e., 550 °C, provided the best confirmation with minimum error value than the other comparative methods. It is clear that by increasing the temperature value, the error value for all of the methods was increased. Because the above values were achieved after 25 runs as mean value of each algorithm, their standard deviation value should be also considered to show their consistency during different independent runs. The standard deviation results of the studied case under different temperature value and 3 atm constant pressure value are given in Figure 3.

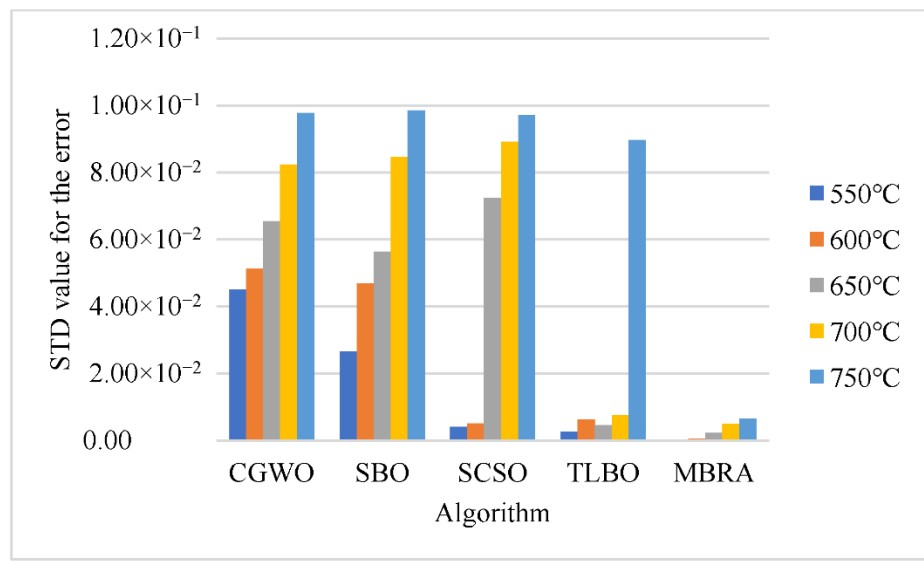

**Figure 3.** Standard deviation results of the studied case under different temperature value and 3 atm constant pressure value.

As can be inferred from the results of Table 6 and Figure 3, there is a minimum error value for all of the metaheuristic algorithms which shows their efficient ability in model parameter estimation of the SOFC. However, the proposed DBRA algorithm with $9.41 \times 10^{-5}$ error in 550 °C provided the minimum value toward the others which shows its higher accuracy than the others. It can also be inferred from Figure 3 that there is an observable difference between the proposed DBRA and the other algorithms in their reliability, which shows the propose method's higher consistency during 25 independent runs. After parameter estimation of the solid oxide fuel cell system, the value of the unknown variables can be achieved and are reported in Table 7.

**Table 7.** Achieved optimal value of the unknown parameters under different temperature conditions.

| Parameters | 550 °C | 600 °C | 650 °C | 700 °C | 750 °C |
|:---:|:---:|:---:|:---:|:---:|:---:|
| $I_{o,a}\,(\mathrm{mA{\cdot}cm^{-2}})$ | 13.96 | 15.50 | 19.81 | 22.68 | 24.41 |
| $I_{o,c}\,(\mathrm{mA{\cdot}cm^{-2}})$ | 7.11 | 7.28 | 7.42 | 7.49 | 7.53 |
| $I_L\,(\mathrm{mA{\cdot}cm^{-2}})$ | 149.73 | 153.35 | 159.66 | 165.85 | 167.26 |
| $E_{oc}\,(\mathrm{V})$ | 1.35 | 1.30 | 1.49 | 1.39 | 1.28 |
| $A\,(\mathrm{V})$ | 0.0448 | 0.045 | 0.047 | 0.049 | 0.051 |
| $B\,(\mathrm{V})$ | 0.046 | 0.049 | 0.054 | 0.062 | 0.077 |
| $R_{ohm}\,(\mathrm{K\Omega{\cdot}cm^{-2}})$ | 0.17 | 0.06 | 0.01 | 0.007 | 0.005 |
| SSE | $9.41 \times 10^{-5}$ | $6.39 \times 10^{-4}$ | $8.60 \times 10^{-4}$ | $1.63 \times 10^{-3}$ | $3.97 \times 10^{-3}$ |
| $R^2$ value | 0.99991 | 0.99985 | 0.99977 | 0.99635 | 0.99664 |
| Accuracy | 99.03 | 97.47 | 97.07 | 95.96 | 93.70 |

As can be inferred from Table 7, the proposed method with $9.41 \times 10^{-5}$ SSE value in 550 °C provides the highest confirmation with the real value and its results get weaker by increasing the temperature value, where in the highest experimented temperature (750 °C), the maximum SSE value ($3.97 \times 10^{-3}$) was achieved. The temperature variations provide a strong upshot on the estimator, i.e., $i$th incrementing of the value of temperature, the density amount of the exchange current for positive and negative is increased, though the voltage is decreased. Moreover, Table 7 shows that $R^2$ values for both training and testing data were extremely close to 1.00. As a result, we can infer that the proposed approach was flawlessly conducted and could precisely anticipate SOFC voltage with the exception of a few severe border situations. We can also prove the better efficiency of the proposed method from the accuracy (99.03%) results.

Figure 4 indicates how the suggested technique gives a promising confirmation with the empirical data under different temperatures.

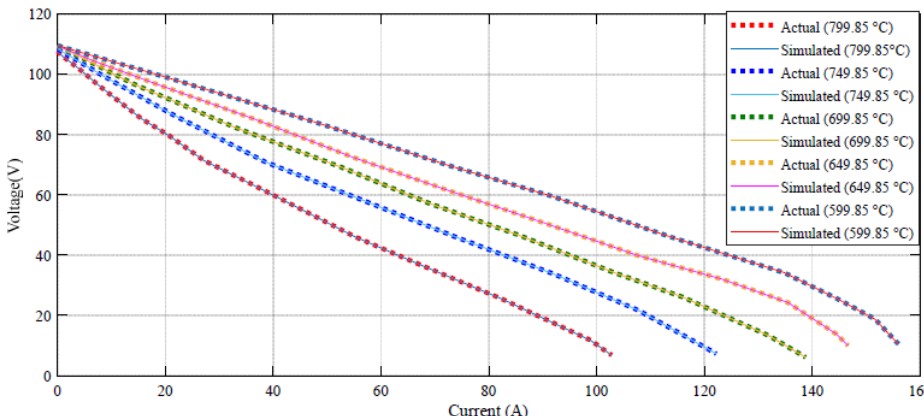

**Figure 4.** Voltage–current profile of the proposed DBRA and its validation with the empirical data.

Based on Figure 4, the error between the DBRA and experimental data is negligible in high values of voltage, such that it cannot be detected in the profile. The error profile of the voltage–current profile during 160 different datapoints are shown in Figure 5.

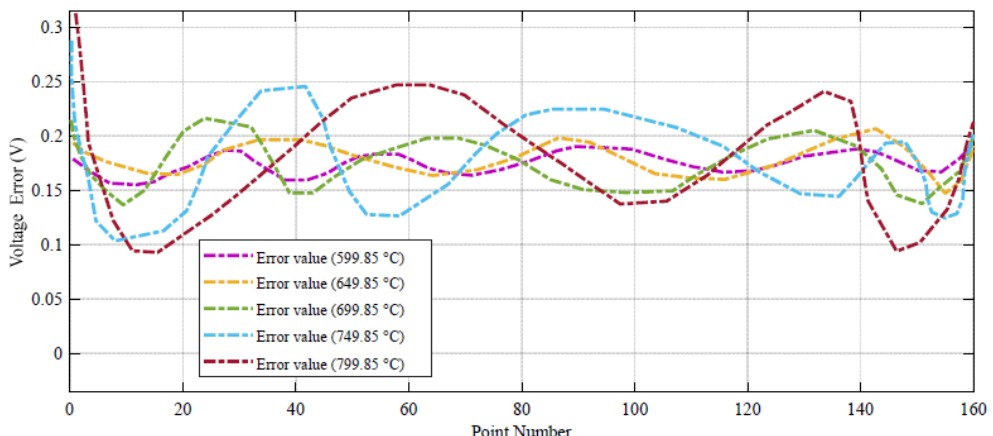

**Figure 5.** Error profile of the voltage–current profile during 160 different datapoints.

As can be seen from Figure 5, the voltage error value in 599.85 °C has the minimum range. For more clarification, the power–current profile of the system during 160 different datapoints are shown in Figure 6.

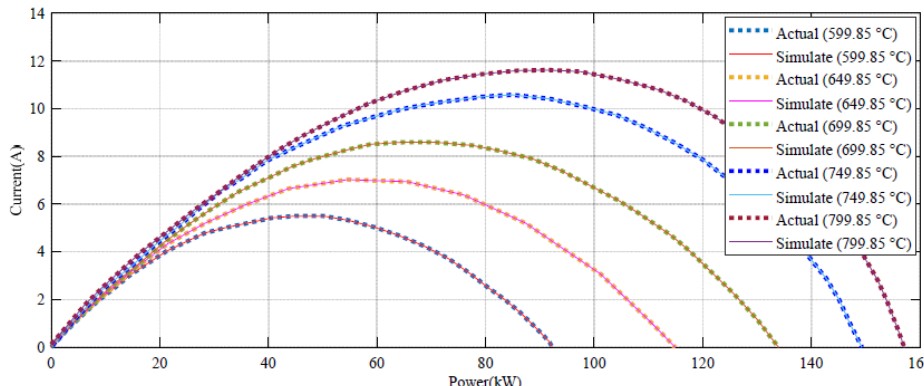

**Figure 6.** Power-current profile of the proposed DBRA and its validation with the empirical data.

Based on Figure 6, the power–current error between the DBRA and empirical data is negligible in high values of power, such that it cannot be detected in the profile. The error profile of the power–current profile during 160 different datapoints is shown in Figure 7.

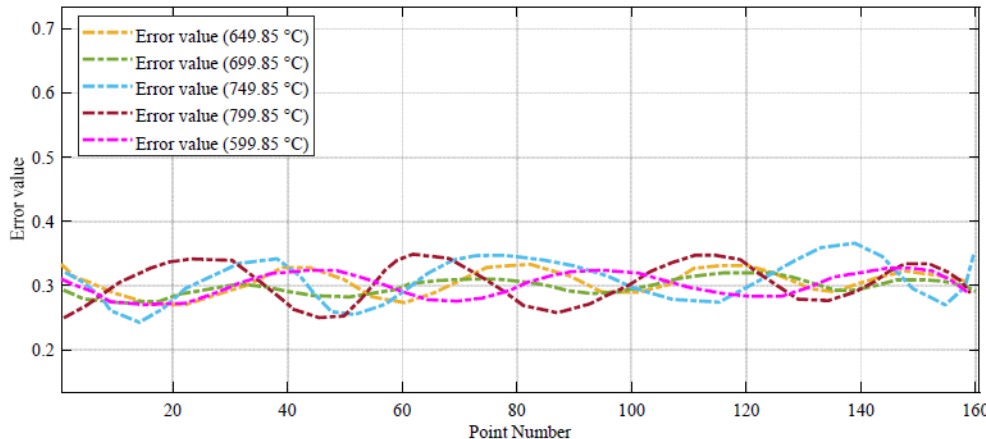

**Figure 7.** Error profile of the power–current profile during 160 different datapoints.

It can be observed from Figure 7 that the presented DBRA has a high validation with the experimental data based on power–current datapoints. In the second step, the system efficiency was analyzed based on pressure variations under constant temperature. Therefore, this calculation indicates how pressure variations affect the estimator. Like the previous analysis, 160 datapoints were also utilized for the assessment. Here, five various pressure values which were 1–5 atm under 750 °C constant temperature value were studied. Table 7 reports the simulation achievements of the presented method under different pressures in comparison with some of the latest optimizers.

Based on Table 8, the error value by the suggested DBRA was minimal in comparison with other newest optimizers, which shows the suggested technique's higher efficiency toward the others. The error curve of the current–voltage of the system under different pressure during 160 different datapoints are shown in Figure 8.

According to Table 8, the lowest error amount was achieved for the proposed DBRA. However, TLBO also had a satisfying error value as the second rank. Furthermore, based on Figure 8, the suggested technique with minimum standard deviation value provided the best confirmation by the experimental data. Table 9 indicates the achieved optimum amount of the undetermined parameters.

**Table 8.** Achieved optimum amount of the undetermined parameters by different algorithms.

| Algorithms | 1 atm | 2 atm | 3 atm | 4 atm | 5 atm |
|---|---|---|---|---|---|
| CGWO [47] | 2.58 | 3.05 | 3.59 | 3.70 | 3.86 |
| SBO [48] | 1.43 | 2.10 | 2.54 | 3.09 | 3.85 |
| SCSO [49] | $4.18 \times 10^{-1}$ | $5.50 \times 10^{-1}$ | $7.25 \times 10^{-1}$ | $9.12 \times 10^{-1}$ | $9.88 \times 10^{-1}$ |
| TLBO [50] | $7.46 \times 10^{-2}$ | $9.80 \times 10^{-2}$ | $1.16 \times 10^{-1}$ | $5.17 \times 10^{-1}$ | $6.94 \times 10^{-1}$ |
| DBRA | $9.43 \times 10^{-3}$ | $4.39 \times 10^{-2}$ | $6.82 \times 10^{-2}$ | $8.17 \times 10^{-2}$ | $9.90 \times 10^{-2}$ |

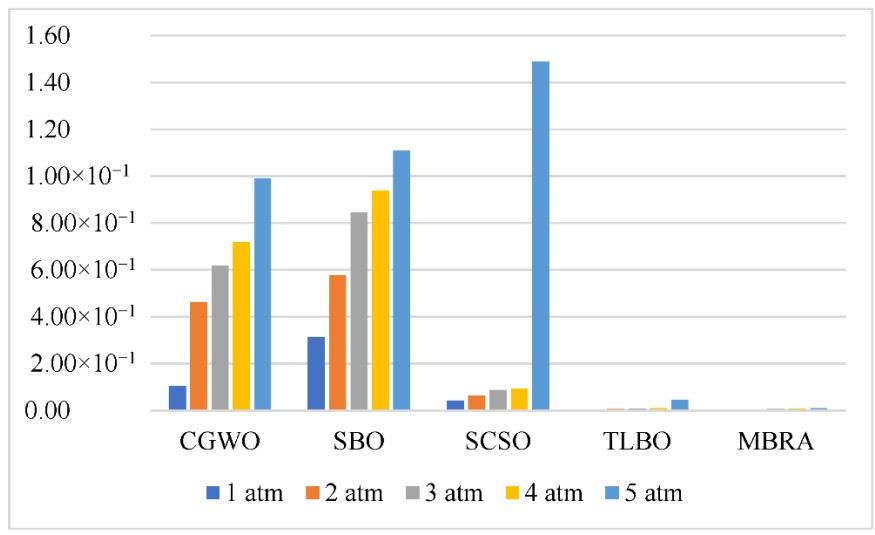

**Figure 8.** Standard deviation results of the studied case under different pressure values and 750 °C constant pressure value.

**Table 9.** Achieved optimum amount of the undetermined parameters by various pressure values.

| Parameters | 1 atm | 2 atm | 3 atm | 4 atm | 5 atm |
|---|---|---|---|---|---|
| $I_{o,a}\left(\mathrm{mA\cdot cm^{-2}}\right)$ | 28.28 | 28.36 | 28.38 | 28.40 | 28.44 |
| $I_{o,c}\left(\mathrm{mA\cdot cm^{-2}}\right)$ | 7.14 | 7.17 | 7.20 | 7.22 | 7.23 |
| $I_L\left(\mathrm{mA\cdot cm^{-2}}\right)$ | 161.42 | 161.45 | 161.52 | 161.53 | 161.56 |
| $E_{oc}(\mathrm{V})$ | 1.18 | 1.22 | 1.27 | 1.35 | 1.50 |
| $A(\mathrm{V})$ | 0.043 | 0.043 | 0.043 | 0.043 | 0.043 |
| $B(\mathrm{V})$ | 0.086 | 0.086 | 0.086 | 0.086 | 0.086 |
| $R_{ohm}\left(\mathrm{K\Omega\cdot cm^{-2}}\right)$ | 0.016 | 0.016 | 0.016 | 0.016 | 0.016 |
| MSE | $1.16 \times 10^{-3}$ | $2.43 \times 10^{-3}$ | $6.95 \times 10^{-3}$ | $8.14 \times 10^{-3}$ | $9.19 \times 10^{-3}$ |
| $R^2$ value | 0.9985 | 0.9949 | 0.9918 | 0.9911 | 0.9904 |
| Accuracy | 97.18 | 95.14 | 91.66 | 90.98 | 90.35 |

It is observed from Table 9, with pressure value increasing, the Nernst voltage value of the mode increased. This indicates the direct relation between pressure value and Nernst voltage value. However, there is no clear elation between the pressure value and the ohmic resistance. Furthermore, Table 9 shows that $R^2$ values for both training and testing data were extremely close to 1.00. As a result, we can infer that the proposed approach was flawlessly conducted and can precisely anticipate SOFC voltage with the exception of a few severe border situations. We can also prove the better efficiency of the proposed method from the accuracy (99.18%) results.

Figure 9 indicates how the suggested technique gives a promising confirmation with the empirical data under different pressure.

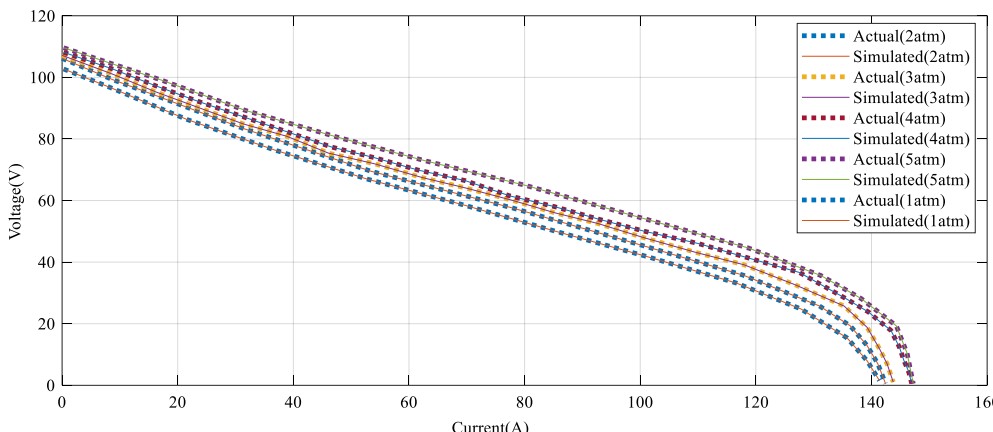

**Figure 9.** Voltage–current profile of the proposed DBRA and its confirmation with the experimental data by various pressure conditions.

According to Figure 9, the error amount between the DBRA and experimental data is negligible in high values of voltage, such that it cannot be detected in the profile. Finally, Figure 10 illustrates how the power–current profile provides a satisfying confirmation with the experimental data by various pressure.

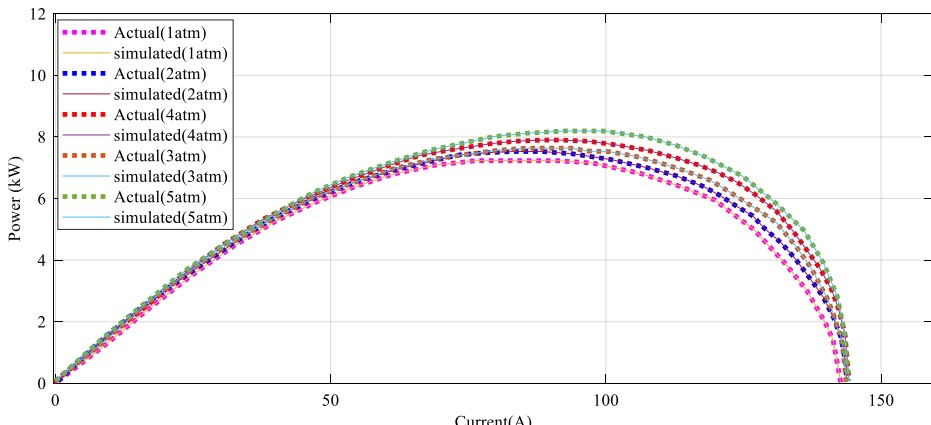

**Figure 10.** Power–current profile of the proposed DBRA and its confirmation with the experimental data by various pressure conditions.

As seen in Figure 10, the power–current error between the DBRA and experimental data under rent pressure values is negligible in high values of power, such that it cannot be detected in the profile.

## 6. Conclusions

The current study proposed a new optimal methodology for efficient identification of the undetermined parameters in the solid oxide fuel cell models. The idea was to lessen the sum of square error between the empirical data and the model output data. The minimum error value in this study was provided with the help of a new developed version of the Battle Royale algorithm to achieve results with higher accuracy and better convergence. The reason for using this version of the Battle Royale algorithm was to attain results with higher accuracy and better convergence. The designed method was then performed on a 96-cell SOFC stack under different temperature and pressure values and its achievements were compared with some of the latest methods to show the proposed

method's effectiveness. The results in the presented method showed high accuracy of the presented developed version of the Battle Royale algorithm; however, using an improved version of the metaheuristic algorithm increases the time complexity of the approach. In future work, we work on proposing a method to simplify the proposed algorithm to get lower time complexity. Moreover, the method will be applied to other fuel cells such as proton-exchange membrane fuel cells to analyze the method's efficiency.

**Author Contributions:** Conceptualization, K.K.A. and M.M.; methodology, A.M.; software, A.M.; validation, A.M.; formal analysis, N.G. and M.G.; investigation, N.G.; resources, N.G.; data curation, A.K.; writing—original draft preparation, M.M.; writing—review and editing, A.K.; visualization, N.G.; supervision, N.G.; project administration, M.G.; funding acquisition, M.M. All authors have read and agreed to the published version of the manuscript.

**Funding:** This research received no external funding.

**Institutional Review Board Statement:** Not applicable.

**Informed Consent Statement:** Not applicable.

**Data Availability Statement:** Not applicable.

**Conflicts of Interest:** The authors declare no conflict of interest.

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
