# Peer review of "Developed Design of Battle Royale Optimizer for the Optimum Identification of Solid Oxide Fuel Cell"

_sustainability, doi:10.3390/su14169882_

Round 1
Reviewer 1 Report
In this work, the author proposed a technique applied to a 96-cell SOFC stack under different 21 temperature and pressure values, and its achievements are put in comparison with several different 22 latest methods to show the proposed method’s efficiency. Some of the minor points need s to be changed before publication:
1. The novelty of this work should be clearly explained in the last paragraph of the introduction section with elaboration.
2. In sections 3 and 4 authors should cite appropriate references related to the explanation and equations.
Author Response
|
Reviewer #1 |
In this work, the author proposed a technique applied to a 96-cell SOFC stack under different temperature and pressure values, and its achievements are put in comparison with several different latest methods to show the proposed method’s efficiency. Some of the minor points need s to be changed before publication:
- The novelty of this work should be clearly explained in the last paragraph of the introduction section with elaboration.
Answer: Dear reviewer, thank you for your constructive recommendation. The novelty of the work has been clearly explained in the last paragraph.
- In sections 3 and 4 authors should cite appropriate references related to the explanation and equations.
Answer: Thank you reviewer. The related references have been cited at the parts to explain the algorithm and equations.
Reviewer 2 Report
Keyvan et al. “Developed design of Battle Royal Optimizer for the Optimum Identification of Solid Oxide Fuel Cell”, this is work is interesting, I recommended minor comments before acceptance.
The introduction should be revised in the standard format of the Journal.
The equation should be arranged in the standard format.
The abstract should be revised and more attractive.
Conclusions should be revised.
All figures should be revised in a standard format.
Author Response
Keyvan et al. “Developed design of Battle Royal Optimizer for the Optimum Identification of Solid Oxide Fuel Cell”, this is work is interesting, I recommended minor comments before acceptance.
The introduction should be revised in the standard format of the Journal.
Answer: Thank you reviewer. The introduction section has been improved based on your recommendation.
The equation should be arranged in the standard format.
Answer: Dear reviewer, thanks to your consideration, we have corrected and updated equations based on the standard format.
The abstract should be revised and more attractive.
Answer: Thank you reviewer. More improvements have been done to the abstract section.
Conclusions should be revised.
Answer: Dear reviewer. The conclusions section has been improved based on your recommendation.
All figures should be revised in a standard format.
Answer: Thank you for the consideration. All of the figures have been replotted based on “.emf” version to provide high quality drawings.
Reviewer 3 Report
Review for
sustainability-1790601
Developed design of Battle Royal Optimizer for the Optimum Identification of Solid Oxide Fuel Cell
Very interesting work and full of practical values. Overall, this article is in good shape, background description is good. I think, nonetheless, that the manuscript could be improved if the authors could address the comments and recommendations I listed below.
According to your description in the Introduction part, are there any similar optimization algorithms worked by others? Albeit you have a solid introduction, you may need some background information about optimization algorithms.
In figure 1. What are the red, pink, and yellow circle means? Please label them as H, O, e-.....
Better highlight novelty in the study.
The data and analyses should be better presented. Add more discussion on the results. Add comparisons with similar methods.
According to your figs 6 & 9 & 10, it looks like your simulated date perfectly matches the actual data. Will you be able to provide the R2? Additionally, you should state what is the accuracy of your simulation? Like XX% accurate.
The conclusion part is not an introduction, this part is wordy, and please concise it. It would be best if you only covered the conclusion based on your findings, rather than repeating the introduction's content. The authors will have to demonstrate the impact and insights of the research. The authors need to clearly provide several solid future research directions. Clearly state your unique research contributions in the conclusion section. Add limitations of the study.
Author Response
Very interesting work and full of practical values. Overall, this article is in good shape, background description is good. I think, nonetheless, that the manuscript could be improved if the authors could address the comments and recommendations I listed below.
According to your description in the Introduction part, are there any similar optimization algorithms worked by others? Albeit you have a solid introduction, you may need some background information about optimization algorithms.
Answer: Dear reviewer, thank you so much for your recommendation. We provided more details in the introduction in the updated paper.
In figure 1. What are the red, pink, and yellow circle means? Please label them as H, O, e-.....
Answer: Dear reviewer, the figure has been drawn again with more explanation.
Better highlight novelty in the study.
Answer: Thank you reviewer, the novelties of the paper are better highlighted at the end of the paper.
The data and analyses should be better presented. Add more discussion on the results. Add comparisons with similar methods.
Answer: Thank you reviewer. We have improved the paper according to the advises that you presented.
According to your figs 6 & 9 & 10, it looks like your simulated date perfectly matches the actual data. Will you be able to provide the R2? Additionally, you should state what is the accuracy of your simulation? Like XX% accurate.
Answer: Thank you reviewer. The recommended analysis are added to the updated paper and discussed under them.
The conclusion part is not an introduction, this part is wordy, and please concise it. It would be best if you only covered the conclusion based on your findings, rather than repeating the introduction's content. The authors will have to demonstrate the impact and insights of the research. The authors need to clearly provide several solid future research directions. Clearly state your unique research contributions in the conclusion section. Add limitations of the study
Answer: Dear reviewer, thank you for your constructive comments. The conclusions section has been improved completely based on our recommendations.
Round 2
Reviewer 3 Report
After checking the draft of the response to the comments, and the corresponding revisions in the revised manuscript, I found that the authors have accomplished the recommended revision to address all my concerns.
This article is good to go.